# Rapid mechanochemical synthesis of polyanionic cathode with improved electrochemical performance for Na-ion batteries

Xing Shen [1,2], Quan Zhou[3], Miao Han[3], Xingguo Qi[3], Bo Li[1], Qiangqiang Zhang[3], Junmei Zhao [1,2,4✉], Chao Yang [1], Huizhou Liu[1] & Yong-Sheng Hu[3✉]

Na-ion batteries have been considered promising candidates for stationary energy storage. However, their wide application is hindered by issues such as high cost and insufficient electrochemical performance, particularly for cathode materials. Here, we report a solvent-free mechanochemical protocol for the in-situ fabrication of sodium vanadium fluorophosphates. Benefiting from the nano-crystallization features and extra Na-storage sites achieved in the synthesis process, the as-prepared carbon-coated $Na_3(VOPO_4)_2F$ nanocomposite exhibits capacity of 142 mAh g$^{-1}$ at 0.1C, higher than its theoretical capacity (130 mAh g$^{-1}$). Moreover, a scaled synthesis with 2 kg of product was conducted and 26650-prototype cells were demonstrated to proof the electrochemical performance. We expect our findings to mark an important step in the industrial application of sodium vanadium fluorophosphates for Na-ion batteries.

---

[1] CAS Key Laboratory of Green Process and Engineering, State Key Laboratory of Biochemical Engineering, Institute of Process Engineering, Chinese Academy of Sciences, Beijing, China. [2] School of Chemical Engineering, University of Chinese Academy of Sciences, Beijing, China. [3] Key Laboratory for Renewable Energy, Beijing Key Laboratory for New Energy Materials and Devices, Beijing National Laboratory for Condensed Matter Physics, Institute of Physics, Chinese Academy of Sciences, Beijing, China. [4] Innovation Academy for Green Manufacture, Chinese Academy of Sciences, Beijing, China. ✉email: jmzhao@ipe.ac.cn; yshu@iphy.ac.cn

Establishing diverse and renewable energy storage systems has attracted wide attention with respect to the grid connection of clean energy sources and achieving sustainable resource utilization. Electric power equipment using batteries is considered among the most promising candidates for future technology in advanced power systems. Over the past few decades, rechargeable batteries have been widely studied and are expected to expedite the development of portable electronic equipment and hybrid or pure electric vehicles[1,2]. Although Li-ion batteries (LIBs) have achieved successful commercialization and conquered the current energy storage market based on their high energy density and long lifespan, the uneven distribution and limited quantity of natural lithium resources make it difficult to meet the ever-increasing demand for lithium, leading to significant cost growth. Hence, there is an urgent need to develop low-cost alternatives with high reserves. Na-ion batteries (NIBs) are promising candidates due to similar working principle as the LIBs concerning the chemical properties and intercalation features. More importantly, the widespread distribution of sodium makes NIBs a cost-effective alternative to LIBs, in particular for large-scale grid storage applications[3–5]. Substantial efforts have been devoted to the development and optimization of low-cost and high-performance electrode materials, in particular cathode materials[6–10], to produce commercially viable NIBs.

Polyanionic compounds, in particular phosphate or fluorophosphate phases, possess a stable three-dimensional host framework structure, owing to strong phosphate-metal bonds, resulting in thermal stability and long lifespans[11–15]. Among them, sodium vanadium fluorophosphates $Na_3(VO_{1-x}PO_4)_2F_{1+2x}$ $(0 \leq x \leq 1$, NVPFs) have been extensively investigated, owing to their high theoretical capacity and working voltage[16–19]. However, the high preparation cost and moderate electrochemical performance limit their large-scale industrial production. The synthesis of pure NVPFs based on the high-temperature solid-state method has been widely investigated[20–22]. This method undoubtedly increases the cost of an electrochemical storage system, but it often results in impurities due to V and F volatilization during high-temperature processing. Therefore, different types of solvothermal and hydrothermal strategies have been employed to prepare NVPFs, to gradually reduce the synthesis temperature[23–28]. Very recently, large-scale room-temperature synthesis was realized through a strategy based on the integration of extraction–separation and material preparation[29]. However, solution-based reactions are usually limited by the solubility, multiple parameters, and slower reaction rate of liquid–solid heterogeneous reactions. Compared to traditional synthetic methods, mechanochemistry is an efficient, quantitative, and solvent-free synthesis process[30,31]. Nowadays, mechanochemical synthesis based on high-energy ball milling (HEBM) has made great strides in the rapid preparation of cathode materials with beneficial high-performance properties[32–35].

In this work, a rapid and solvent-free mechanochemical synthesis of NVPFs via HEBM was investigated. $Na_3(VOPO_4)_2F$ (NVOPF) nanoparticles with excellent rate capability and superior cycling stability can be easily obtained, and carbon-coated NVOPF/Ketjen black (KB) nanocomposites can be in situ-constructed. In particular, the NVOPF/8%KB composite delivers a discharge capacity of 142.2 mAh g$^{-1}$ at 0.1C, which exceeds the theoretical value and exhibits excellent cycling performance with 98% capacity retention after 10,000 cycles at 20C. High structural reversibility was revealed by in situ X-ray diffraction (XRD), ex situ solid-state $^{23}$Na nuclear magnetic resonance (NMR), ultraviolet-visible spectrophotometer (UV-vis), and Raman spectroscopy. Moreover, a scalable synthesis of NVOPF/KBs is also demonstrated with a 26,650 prototype full cell exhibiting superior cycling performance over 100 cycles. The novel synthesis and optimized strategy are of vital importance for the industrial-scale production and further development of NVPFs.

## Results

**Synthesis of NVPFs via HEBM.** Different reaction conditions are required for the successful preparation of phase-pure NVPFs from different vanadium sources[36]. In this work, different vanadium sources, including trivalent (vanadium (III) trichloride ($VCl_3$), vanadium (III) acetylacetonate ($V(C_5H_7O_2)_3$ (abbreviated as $V(acac)_3$), and vanadium trioxide ($V_2O_3$)), tetravalent (vanadyl sulfate ($VOSO_4 \cdot xH_2O$), vanadyl (IV) acetylacetonate (VO($C_5H_7O_2$)$_2$ (abbreiated as VO(acac)$_2$), and vanadium dioxide ($VO_2$)), and pentavalent (ammonium metavanadate ($NH_4VO_3$), sodium metavanadate ($NaVO_3$), sodium orthovanadate dodecahydrate ($Na_3VO_4 \cdot 12H_2O$), and vanadium pentoxide ($V_2O_5$)), were investigated as starting materials to obtain pure NVPFs via HEBM. The schematic illustration of the synthetic process is shown in Fig. 1. The powder acquired from the reaction was washed with deionized water and the product was collected. Supplementary Fig. 1 shows the XRD patterns of the successful preparation derived from ten different precursors, which are well indexed to the standard powder diffraction file card of $Na_3(V-PO_4)_2F_3$ (JCPDS 01-089-8485) or NVOPF (JCPDS 97-041-1950); they are similar except for small differences in peak deviation and intensity, suggesting that all selected vanadium sources can be successfully applied to synthesizing NVPFs by adjusting the corresponding phosphorus sources and reactant molar ratios. The corresponding reaction results are summarized in Supplementary Table 1. As can be seen, both organic and inorganic vanadium salts can react with sodium dihydrogen phosphate dihydrate ($NaH_2PO_4 \cdot 2H_2O$) via HEBM. However, vanadium

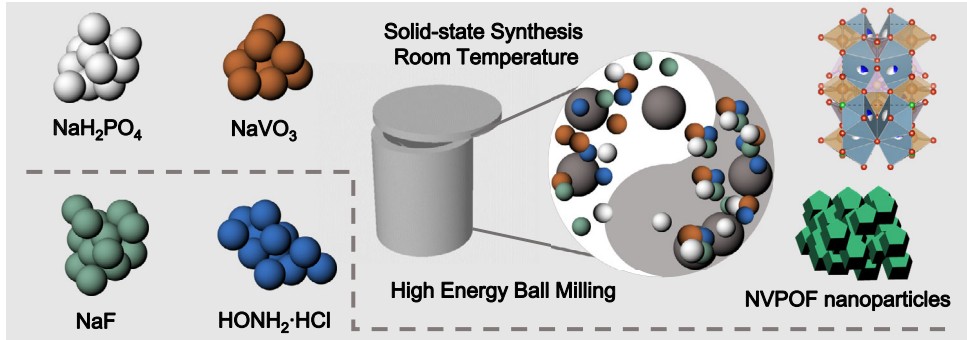

**Fig. 1 The mechanochemical synthesis of Na₃(VOPO₄)₂F nano-particles starting from NaVO₃.** Schematic illustration of the typical synthetic process of NVPFs.

oxides require the stronger acid $H_3PO_4$ (phosphoric acid) to prepare the phase-pure target product.

The pH values of the filtrate ranged between 2.85 and 5.86, depending on the vanadium source, which is almost consistent with the pH range proposed for the solution-based strategy[36]. The ratios of reactants are slightly higher than the stoichiometric ratio to guarantee the conversion of the high-value vanadium source. The UV-vis spectra were employed to clarify the valence of V in the above-mentioned products, as shown in Supplementary Fig. 2. It was observed that the trivalent vanadium product $Na_3(VPO_4)_2F_3$ can only be synthesized by a trivalent vanadium source, whereas NVOPF can be obtained from tetravalent or pentavalent vanadium sources. Successful implementation with various vanadium sources confirmed the universality of the mechanochemical strategy for NVPFs and this scalable synthetic method will undoubtedly promote the industrial applicability of NVPFs.

The manufacturing cost is greatly affected by the vanadium source used. Costs of typical vanadium sources are listed in Supplementary Table 2. Vanadium products made from vanadium-containing natural mineral, vanadium titano-magnetite, or vanadium slags are normally pentavalent. Therefore, inexpensive pentavalent vanadium sources are more suitable for scale-up operations. The cost of NVPFs calculated assuming a 100% yield and the in-time prices of all other raw materials are listed in Supplementary Table 3; the product made from $NaVO_3$ is the least expensive. As shown in Supplementary Fig. 1, the powder obtained from $NaVO_3$ exhibited the best crystallinity. Next, NVOPF synthesized by taking $NaVO_3$ as the vanadium source was optimized and characterized; hydroxylamine was the reductant used to reduce V(V) to V(IV)[29,37]. The chemical reaction equation is similar to that of the reported room-temperature synthesis[29].

The proportion of raw materials was optimized to achieve the maximum yield. The corresponding yield with different dosage of raw materials was calculated from the residual vanadium in the filtrate after the byproducts were washed away, as shown in Supplementary Table 4. The byproducts include NaCl, $NH_4H_2PO_4$, etc., as seen in the XRD pattern of as-synthesized samples without washing (Supplementary Fig. 3). The objective products prepared from different ratios of V, P, and F exhibit identical diffraction peaks, which are in good agreement with the standard card (JCPDS 97-041-1950), as seen in Supplementary Fig. 4. The results show that the optimal ratio is 1 : 1.5 : 1 for V : P : F source, owing to its best crystallinity and a high yield of 94%. In addition, the effects of reaction time on the XRD pattern and product yield were investigated, as shown in Supplementary Fig. 5. It can be concluded that the target product NVOPF began to form in the first 5 min and showed excellent crystallinity at 15 min. The optimized time for the HEBM reaction was found to be 30 min with a high yield of 94%. Products present homogeneous nanoparticles with some agglomeration, as shown in Supplementary Fig. 6. This method, with rapid synthesis in 30 min, is more efficient than other synthetic methods reported thus far.

The powder XRD pattern and Rietveld-refined results of the as-prepared NVOPF are shown in Fig. 2a. A Rietveld structural refinement conducted based on $I4/mmm$ space (tetragonal structural mode) is in good accordance with the cell parameters of similar compounds without impurities and with good reliability factor values (Rwp = 7.43%, Rp = 8.51%). The broadened peaks indicate nanocrystalline features. The average grain size calculated using the Scherrer formula is ~5 nm. The crystallographic and Rietveld refinement data of the NVOPF compound are listed in Supplementary Tables 5 and 6. The unit cell parameters are $a = b = 6.3884$ Å and $c = 10.6582$ Å, which is in good agreement with previously reported values[29,38]. The schematic diagram of the NVOPF tetragonal structure is given in Fig. 2b. The crystal structure can be described in terms of $[V_2O_{10}F]$ dioctahedra and $[PO_4]$ tetrahedra. The oxygen sharing of dioctahedra and tetrahedra leads to the formation of channels along the $a$ and $b$ axis with $Na^+$ located in the tunnel sites. As shown in Fig. 2c, d, NVOPF displays a typical particle size of 30 nm. Furthermore, the high-resolution transmission electron microscope (HRTEM) image presents an interplanar spacing of 4.425 Å in the single particle, which matches well with the (110) plane and indicates the high crystallinity of the as-prepared NVOPF compound.

**In situ construction of the NVOPF/KB nanocomposite with superior cycling stability**. The Na-storage performances of the as-prepared NVOPF nanoparticles were evaluated in half-cells. The charge–discharge curves are plotted at a gradual ascending current rate of 0.1C to 15C in Fig. 3a. The coin cells were tested in the range of 2.5–4.2 V vs. Na/Na$^+$. At the current rate of 0.1C, the charge–discharge curve exhibits two typical plateaus at 4.10/4.05 and 3.65/3.60 V, respectively. Reversible capacities of 120.7, 120.2, 119.7, 118.5, 115.2, 101.2, and 96.5 mAh g$^{-1}$ are achieved at rates of 0.1, 0.2, 0.5, 1, 2, 10, and 15C, respectively. All charge–discharge profiles show two similar responsive platforms, except for the larger hysteresis with increasing current rate. This could be ascribed to the intrinsic poor electron conductivity of polyanionic compounds, which can be alleviated by regulating the charge distribution during electron transfer in electrode materials, such as by carbon coating or mixing[39].

KB is widely used to improve the electronic conductivity of cathode materials because of its unique branch-chain structure and abundant conductive contact points[39–41]. To further enhance the Na-storage performance of NVOPF, we propose the in-situ construction of the NVOPF/C composite via HEBM with different contents of KB. The exact amount of KB was analyzed using an element analyzer, as shown in Supplementary Table 7. Diffraction peak intensities become weaker with increasing KB amounts for the as-prepared NVOPF/KB samples, indicating reduced crystallization in the presence of KB, as seen in Supplementary Fig. 7. The charge–discharge curves for NVOPF with increasing contents of KB at 0.1C are shown in the inset part of Fig. 3b. The initial discharge capacities of the samples at 0.1C are 122.2, 126.8, 142.2, 141.9, and 134.8 mAh g$^{-1}$ for bare NVOPF, NVOPF/6%KB, NVOPF/8%KB, NVOPF/10%KB, and NVOPF/12%KB, respectively. The moderate introduction of KB can build up a well-organized NVOPF/KB matrix and should improve the electrical conductivity of active materials. However, extra KB would cause severe aggregation to reduce the utilization of active materials. The electrode was also tested for rate capability at current rates of 0.1–20C, as depicted in Fig. 3b. It is observed that the rate capability is greatly improved in the presence of KB. In particular, considerable reversible discharge capacities of 103.1, 112.8, 113.2, and 109.8 mAh g$^{-1}$ are recorded at 20C for NVOPF/6%KB, NVOPF/8%KB, NVOPF/10%KB, and NVOPF/12%KB, respectively, whereas it is only 84.8 mAh g$^{-1}$ for bare NVOPF. In addition, NVOPF/KB also exhibits ultralong cycling stability, as displayed in Supplementary Fig. 8. To the best of our knowledge, NVOPF/8%KB is much superior to the other reported NVPFs samples with respect to discharge capacity and rate capability, as summarized in Supplementary Table 8. Moreover, the current material presents the highest working voltage in NIBs compared with other kinds of polyanionic compounds, layered oxide and Prussian blue analogs[42–46], as shown in Fig. 3c. Also, it delivers a 555 Wh kg$^{-1}$ of theoretical energy density based on cathode, which is very close to the highest energy density report about the polyanionic compound $Na_4MnCr(PO_4)_3$ (566 Wh kg$^{-1}$)[47]. It is worth noting that the excess capacity beyond

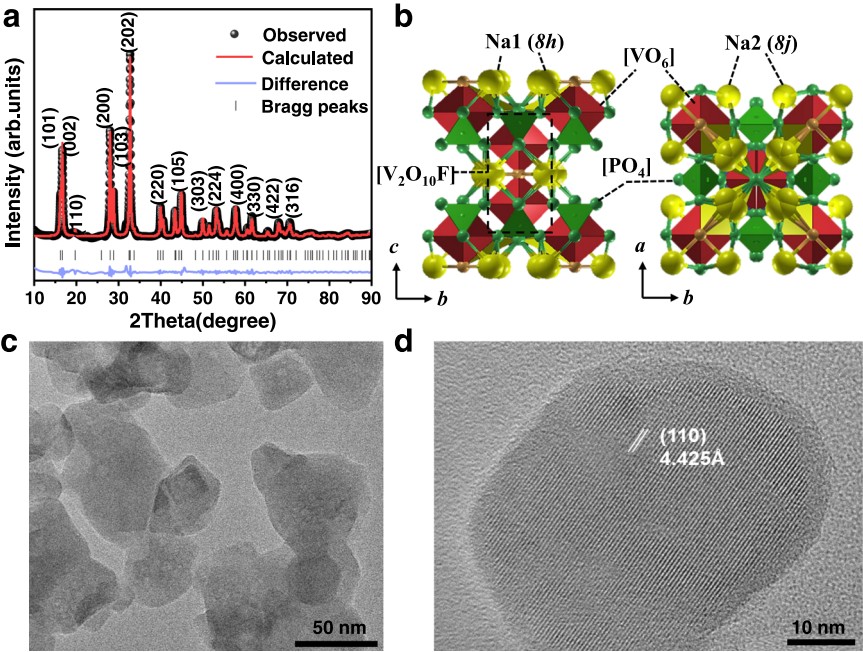

**Fig. 2 Crystal and morphological structure characterizations. a** Rietveld-refined profiles of as-synthesized NVOPF. **b** Schematic diagram of NVOPF structure, **c** TEM, and **d** HRTEM image of as-prepared NVOPF.

the theoretical value was observed for NVOPF/KBs (8–12%) at 0.1C, which can be ascribed to the formation of extra Na-storage sites in the in situ-constructed interface of NVOPF/KB nanocomposite. Taking NVOPF/8%KB as an example, we reconfirmed the discharged capacity with different batches of samples at 0.1C and the average capacity is 141.9 mAh g$^{-1}$, as seen in Supplementary Fig. 9a. The KB with and without ball milling was discharged to 2.5 V, to reveal the corresponding Na-storage properties. It can be found that the discharged capacity of KB with and without ball milling are 3.5 and 0.7 mAh g$^{-1}$, respectively (Supplementary Fig. 9b, c). The X-ray photoelectron spectroscopy was employed to observe the chemical state of discharged KB electrodes, as shown in Supplementary Fig. 9d. No Na signal was detected in the discharged KB without ball milling, but an obvious Na 1s peak located at ~1069.2 eV can be found in the discharged KB with ball milling, which confirms the binding effect between Na and KB. Based on this, we collected the Na 1s X-ray photoelectron spectroscopy (XPS) data of NVOPF and NVOPF/8%KB electrodes with varied charge/discharge states. As shown in Supplementary Fig. 10a, there is no obvious change in the Na peak for NVOPF during the first charge–discharge process, declaring only Na$^+$ of the crystal structure migrates in the electrochemical reaction. Whereas for NVOPF/8%KB, the extra Na-C peak can be detected in the pristine electrode, implying that Na-C bond forms in the ball-milled process. After charged to 4.2 V, the Na-C peak disappears as the extraction of Na$^+$ from the NVOPF/8%KB species. When the NVOPF/8%KB electrode was discharged to 2.5 V, an extra Na peak emerged at 1071.7 eV, which denotes the formation of new Na-site at this state, as seen from Supplementary Fig. 10b. The new Na-site at 1071.7 eV is not derived from the bulk of NVOPF or KB, but could be derived from the interfacial between NVOPF and KB. To better describe the interfacial storage mechanism, the schematic representation is shown in Supplementary Fig. 11. In the interface between NVOPF and KB, Na$^+$ can be accommo-dated at the boundary of NVOPF side, while the electrons are restricted to the KB side. For the combination of NVOPF and KB, the stored Na$^+$ and e$^-$ act as a bridge during the charge–discharge process; thus, an interfacial Na storage can be

expected. This phenomenon has also been reported for nano-ionics in the context of lithium batteries, such as LiFePO$_4$[48–50]. Except for the over-capacity capability, NVOPF/8%KB can maintain a discharge capacity of 110.5 mAh g$^{-1}$ after 10,000 cycles at 20C, which is 98% of the initial discharge capacity, with a fade rate of only 0.0002 mAh g$^{-1}$ per cycle, as shown in Fig. 3d, whereas the capacity retention is only 72% under the same charge–discharge conditions for bare NVOPF and the corre-sponding charge–discharge curves were plotted in Fig. 3e, f, with a moderate loading amount of 6 mg cm$^{-2}$. The NVOPF/8%KB cathode could keep its original crystal structure even suffered from the violent decomposition of electrolyte, as the post-mortem analysis shown in Supplementary Fig. 12. Based on this, compact coin full cell was fabricated to verify the match feasibility of NVOPF/KB and hard carbon (HC). As seen in Fig. 3g, a light-emitting diode lamp was lit successfully by one coin cell and was worked normally for several minutes. The full cell presents a 94.3 mAh g$^{-1}$ discharge capacity at the current rate of 0.2C (Supplementary Fig. 13a). As shown in Supplementary Fig. 13b, the capacities delivered along with rate capability in full-cell configuration with HC were also tested from 0.5C to 10C, and they are 86.5, 73.9, 66.9, 57.5, and 48.9 mAh g$^{-1}$ for 0.5, 1, 2, 5, and 10C, respectively. Besides, the cell exhibited a capacity retention of 90.7% after 100 cycles at the current rate of 2C, which is much superior to the reported full-cell systems, as summarized in the Ragone plots in Supplementary Fig. 13c.

The scanning electron microscopy (SEM), TEM, and corre-sponding elemental mapping images of NVOPF/8%KB are shown in Supplementary Fig. 14. This demonstrates the homogeneous elemental distribution in these nanocomposites and the cross-linking between NVOPF and KB. The as-prepared samples thus display a stronger affinity between carbon and active material than those from the general two-step synthesis[39,41,49]. Further-more, detailed structural information can be obtained by HRTEM, as shown in Supplementary Fig. 15. The NVOPF/8% KB sample shows characteristic spacings of 0.514, 0.445, and 0.279 nm for the (002), (110), and (022) lattice planes of the tetragonal structure. In contrast to bare NVOPF (Fig. 2d), a more disordered structure with smaller particle size was formed for

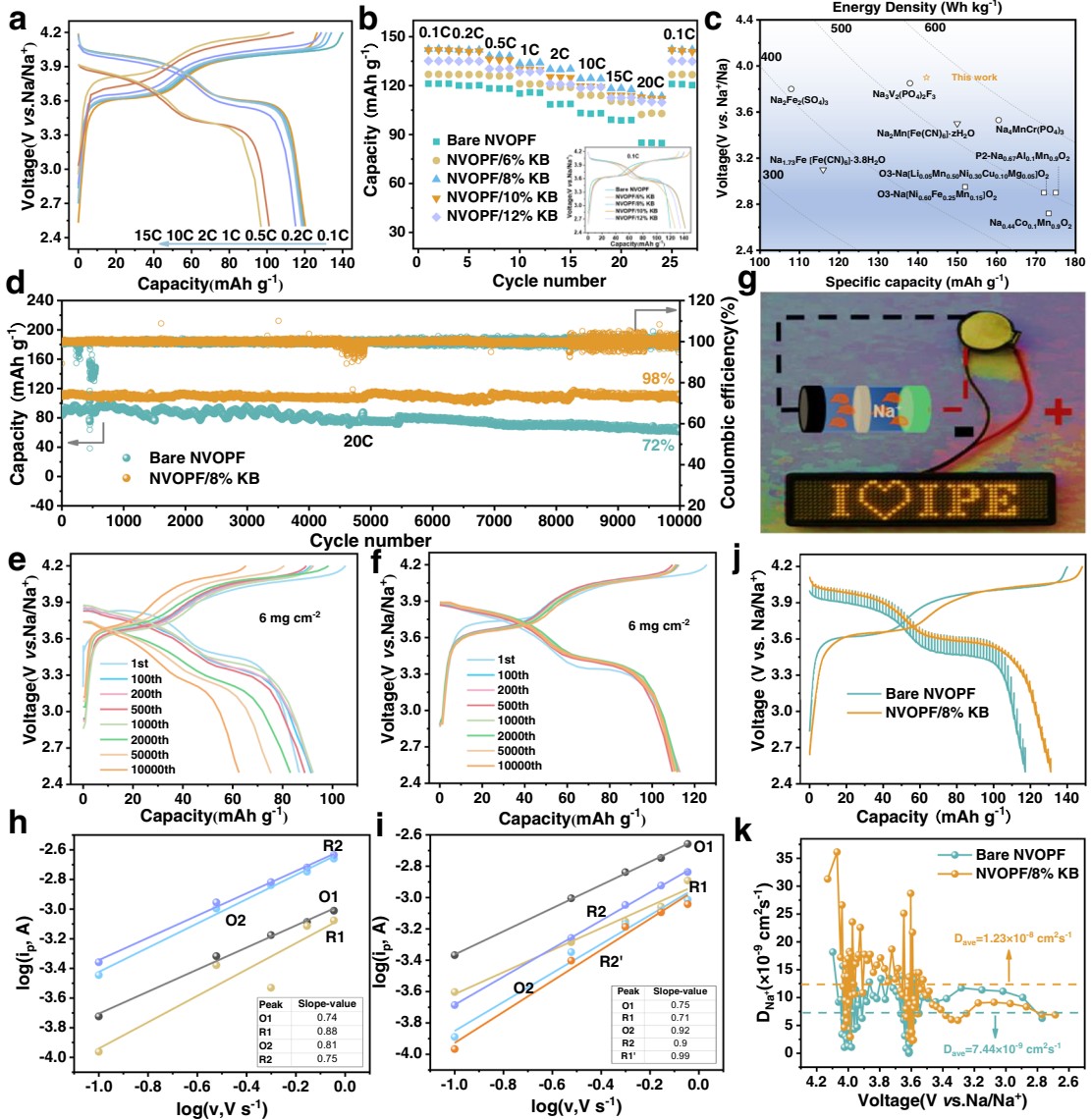

**Fig. 3 Electrochemical properties and Na ion diffusion kinetics. a** The rate capability of NVOPF electrode at various rates ranging from 0.1C to 15C. **b** Rate capability of NVOPF with different KB contents at rates ranging from 0.1C to 20C (inset: the typical charge and discharge curves of NVOPF with different KB contents at 0.1C). **c** Relationships between the working voltage, specific capacity, and energy density in half-cells for NIBs. **d** The cycling performance of bare NVOPF and NVOPF/8% KB electrode at 20C. **e** The charge–discharge curves of bare NVOPF cathode at 20C. **f** The charge–discharge curves of NVOPF/8%KB cathode at 20C. **g** Photograph of LED lamps lighted by the coin full cell. **h** Linear fittings of log ($i_p$) and log (v) for bare NVOPF electrode (inset: the corresponding pseudocapacitive contributions and peak slope values). **i** Linear fittings of log ($i_p$) and log (v) disclosing the relationships between peak currents and scan rates from the corresponding CV curves of NVOPF/8%KB electrode. **j** The GITT curves of bare NVOPF and NVOPF/8%KB. **k** Corresponding evolution of diffusion coefficient of bare NVOPF and NVOPF/8%KB electrodes during the discharging process.

NVOPF/8%KB with the faint lattice fringes in the presence of KB. Besides, Rietveld-refined results based on the power XRD of NVOPF/8%KB composites present a lattice volume of 435.22 Å³, which is a little bit larger than pure NVOPF (434.98 Å³), as shown in Supplementary Fig. 16 and Supplementary Table 9. Therefore, the in situ reaction in the ball-milling process may produce defects or amorphized structure in the interface between NVOPF and KB, which could also benefit to the formation of a more disordered structure. These results suggest that the in situ construction strategy provides a simple but very efficient one-step route for the preparation of carbon-containing samples. The reduced size of NVOPF in the carbon matrix can also effectively shorten the diffusion length of Na⁺ ions in the composite, contributing to faster ion transportation.

To compare the Na⁺ ion diffusion kinetics of bare NVOPF and NVOPF/8%KB, cyclic voltammograms between 2.5 and 4.2 V were recorded at series scanning rates (0.1, 0.3, 0.5, 0.7, and 0.9 mV s⁻¹). Both cyclic voltammetry (CV) curves for bare NVOPF and NVOPF/8%KB at varied scan rates exhibit two main redox peaks at voltages of 3.70 and 4.05 V, indicating excellent reversibility during the charge/discharge processes, as depicted in Supplementary Fig. 17. It is worth noting that the position deviations between the oxidation peaks and corresponding reduction peaks demonstrate increasing polarization as the scanning rate increases. In contrast to bare NVOPF, the distinct reduction peak of NVOPF/8%KB below 3.5 V was split into two mild peaks (3.25 and 3.46 V), as shown in Supplementary Fig. 17b. This phenomenon has also been reported for other

similar polyanion compounds[51]. Typically, the peak current ($i_p$) and scan rate ($v$) follow the equations given below[52,53]:

$$i_p = av^b$$

$$\log\left(i_p\right) = b\log(v) + \log(a)$$

where $a$ and $b$ are adjustable parameters[52,54]. By capturing the peak currents ($i_p$) of the cathodic (O1, O2) and anodic peaks (R1, R2) at every scanning rate ($v$), the log($i_p$) vs. log($v$) plots and the corresponding fitted linear correlations are plotted as shown in Fig. 3h, i. The $b$-values of O1, R1, O2, and R2 are calculated to be 0.74, 0.88, 0.81, and 0.75 for bare NVOPF, respectively; they are 0.75, 0.71, 0.92, and 0.90 for NVOPF/8%KB, respectively, which are >0.5 and close to 1. In particular, the $b$-value of the extra reduction peak R1' for NVOPF/8%KB is 0.99. These values reveal a high pseudocapacitive contribution during the charge and discharge processes of the NVOPF cathodes. It can be rationally deduced that the plateau regions at the O1/R1 peaks belong to a relatively slow Na$^+$ extraction process, whereas the high-potential process dominated by surface-active sites is attributed to a fast charge transfer process. Specifically, the proportions of non-faradic contributions (Supplementary Fig. 18) are about 56.17%, 57.32%, 58.71%, 61.04%, and 63.54%, and 82.97%, 84.51%, 87.34%, 91.36%, and 94.64%, at the scan rates of 0.1, 0.3, 0.5, 0.7, and 0.9 mV s$^{-1}$, for bare NVOPF and NVOPF/8%KB, respectively. The calculation details are shown in Supplementary Fig. 19. This further demonstrates that pseudocapacitive Na-storage dominates the charging/discharging processes, especially at high current rates, confirming the fast-charging ability of NVOPF/8%KB. The superior rate capacities could be related with the increased specific surface areas. As is evident from the nitrogen adsorption–desorption isotherms in Supplementary Fig. 20, the bare NVOPF and NVOPF/8%KB showed a Brunauer–Emmett–Teller (BET) surface area of 30.443 and 44.572 m$^2$ g$^{-1}$, respectively. The pore volume and pore diameter data are listed in Supplementary Table 10. The larger specific area and average pore diameter of NVOPF/8%KB may be beneficial to the infiltration of electrolyte due to the abundance of ionic transmission channels.

Further evidence in favor of the ion transport capacity of NVOPF/KB nanocomposite is provided by galvanostatic intermittent titration technique (GITT) at a testing rate of 0.083C, within the potential range of 2.5–4.2 V during discharging in the initial cycle. As seen in Fig. 3j, the NVOPF/8%KB sample exhibits a smaller overpotential (10–20 mV) than bare NVOPF (130–150 mV) during the discharging process, indicating better kinetics for Na$^+$ diffusion in the NVOPF/8%KB composite. For single titration, greater voltage drop caused by pulses ($\triangle E_\tau$, $E_{A3} - E_{A2}$) and constant current charge–discharge ($\triangle E_s$, $E_{A1} - E_{A4}$) (Supplementary Fig. 21) can be observed for bare NVOPF. Based on the voltage values of four marked characteristic points, corresponding diffusion coefficient values can be calculated in sequence using Supplementary Eqn (2), derived from Fick's second laws of diffusion[55]. The $D_{Na}^+$ values obtained are in an order of magnitude of $10^{-9}$ to $10^{-8}$ cm$^2$ s$^{-1}$, with an explicit decreasing trend upon discharging. Intriguingly, $D_{Na}^+$ fluctuates wildly at plateaus of 3.6 and 4.0 V, which can be attributed to the abundant insertion of sodium ions in these voltage regions[56]. In addition, the calculated Na$^+$ diffusion coefficient shown in Fig. 3k discloses that the average values for $D_{Na}^+$ are about $7.44 \times 10^{-9}$ and $1.23 \times 10^{-8}$ cm$^2$ s$^{-1}$ for bare NVOPF and NVOPF/8%KB, respectively. The electrochemical impedance spectroscopy results also show that the half cell using NVOPF/8%KB as cathode has a lower charge transfer resistance than bare NOVPF, no matter for cells prior to cycling (245.3 Ω vs.

294.4 Ω) or after one cycle (137.3 Ω vs. 147.3 Ω), as seen in Supplementary Fig. 22 and Supplementary Table 11. These results demonstrate that Na diffusion kinetics can be greatly improved through the in situ construction of the NVOPF/KB nanocomposite.

To elucidate the the evolution of crystal structure induced by the de-sodiation/sodiation during the charge–discharge process, in situ XRD tests were performed with a sampling interval of 25 min for the first cycle at a current rate of 0.1C. As seen in Fig. 4a, a 138.2 mAh g$^{-1}$ discharge capacity was obtained for the initial cycle, which is almost identical with the coin cell, proving the high dependability of in situ XRD results. The collected in situ XRD patterns are shown in Fig. 4b. For the fresh electrode, the XRD pattern was similar to the as-synthesized NVOPF/8%KB sample. Detailed structural evolution during cycling was monitored via the evolution of the diffraction peaks, especially for the (200), (103), and (202) planes. For the diffraction of the (200) crystal planes, the peak located at 28.6° weakens gradually and shifts to higher 2θ degrees, and eventually converges at 29.1° with the (103) planes. Meanwhile, the peak of the (103) face shifts toward lower degrees during the charging process to 4.2 V. Similarly, the merging of diffraction peaks was also observed in the case of the other two pairs of adjacent peaks of (006) and (231), and (224) and (125). The isolated diffraction peaks of the (202) and (330) planes exhibit a similar shift to a higher 2θ degree. Figure 4c depicts the changes in $a$, $c$, and cell volume values in the unit cell, showing that the $c$-values increases owing to the increased repulsive force between the pseudolites derived from the extraction of Na ions, whereas $a(b)$ decreases continuously with the oxidation of vanadium from V (IV, 0.058 nm) to V (V, 0.054 nm). Inversely, these peaks are steadily restored to the pre-charging position via the return path of splitting and variation as it discharged to 2.5 V, manifesting the reversible evolution of NVOPF phase stem from its stable but flexible framework. The continuous, reversible movements of all diffraction peaks suggest the characteristics of a solid-solution reaction[18,28,30]. It was observed through calculations that the change in NVOPF/8%KB unit cell volume was as low as 0.47%, which is much smaller than the 5.23% for bare NVOPF calculated from the ex situ XRD patterns shown in Supplementary Fig. 23, indicating an enormously low lattice strain for NVOPF/8%KB during the charging and discharging processes.

To further ascertain the reaction mechanism and local structural evolution, high-resolution $^{23}$Na SSNMR spectra of NVOPF/8%KB electrodes under various charge/discharge states (pristine, half-charged, fully charged, half-discharged, fully discharged) were tested and the results are presented in Fig. 4d. According to the working principle of solid-state NMR, unpaired electron spins can be transferred to Na 3s orbitals through O/F 2p orbitals; Na-V(IV) is expected to show a larger Fermi-contact shift, as it contains one unpaired electron ($t_{2g}^1 e_g^0$), whereas no electron is contained in V(V) ($t_{2g}^0 e_g^0$)[57–59]. For the pristine electrode, a dominant Lorentzian line centered at 75 p.p.m. can be ascribed to Na$^+$ being neighbored with V (IV), which suggests that the Na1 and Na2 sites in the crystal structure are not distinguished here, and share the same isotropic chemical shift[57]. The faint peaks at ~0 p.p.m. and <0 p.p.m. presented in the spectra are attributable to the Na from the diamagnetic Na salts in the cathode electrolyte interphase and electrolyte (NaClO$_4$ in propylene carbonate (PC)), respectively[57]. As the Na-extraction process is accompanied by vanadium oxidation with a decrease of unpaired electrons, the peak centered at 75 p.p.m. (Na-V(IV)) shows an abrupt decline. From half to the fully charged state, the Na-V(IV) peak at 75 p.p.m. greatly declines and the characteristic peak of Na-V(V) at 30 p.p.m. becomes dominant, indicating the changing chemical environment of Na triggered by the oxidation

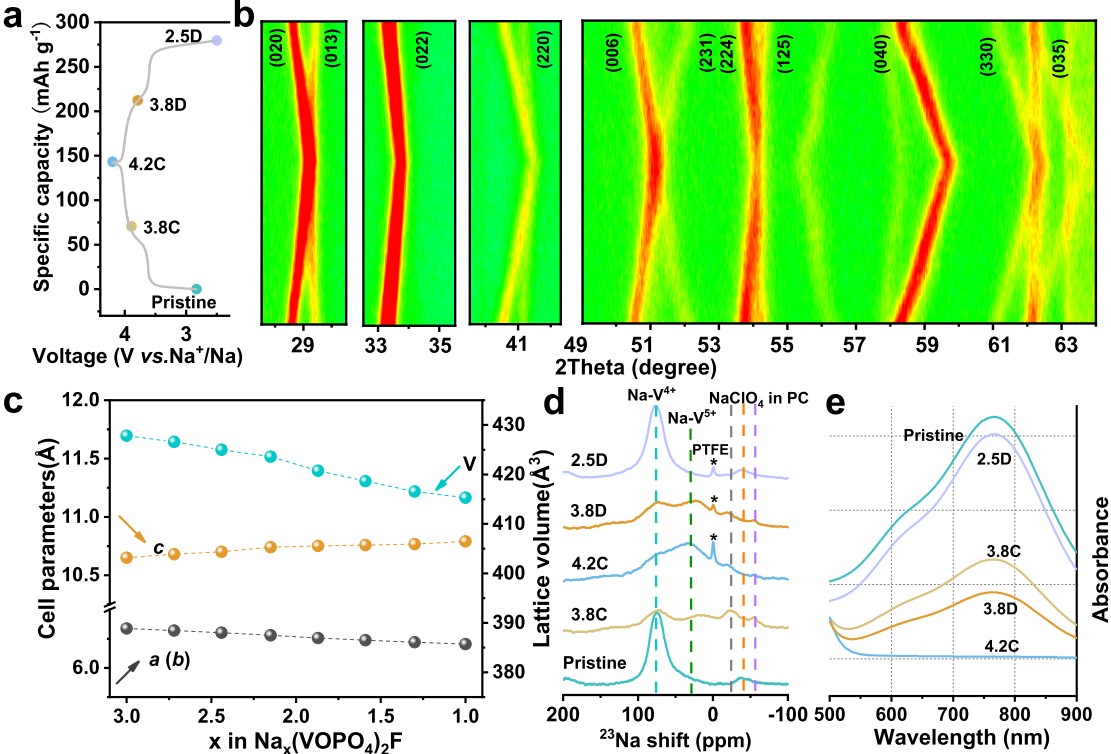

**Fig. 4 Crystal structure evolution of NVOPF/8%KB cathode. a** The charge–discharge curves of NVOPF/8%KB at a current rate of 0.1C in in situ XRD experiment. **b** The in situ XRD patterns. **c** The variation of cell parameters during the charging process. **d** High-resolution $^{23}$Na SSNMR spectra of NVOPF/8%KB electrode under various states. **e** Ex situ UV-vis spectra of the NVOPF/8%KB electrode.

of V(IV). When the electrode was discharged to 3.8 V, it returned to the same state as the half-charged sample, representing the solid-solution reaction from NV(V)OPF to NV(IV)OPF. Compared to the fully charged sample, the V(IV)–V(V) coexistence state (half-charged or half-discharged) samples show a smaller Fermi-contact shift for Na-V(V) due to the exceedingly small difference in ambient vanadium ions. After being fully discharged to 2.5 V, the $^{23}$Na spectrum of the cycled NVOPF/8%KB recovers to a similar state as the pristine sample, further confirming the excellent reversibility of the local structural evolution process, which is consistent with the in situ XRD results.

In addition, ex situ UV-vis spectra were obtained to identify vanadium oxidation at different stages during the initial cycle. Based on our previous work[29], the standard solution of V(IV) has two significant peaks at 630 and 765 nm; the former is a shoulder peak and the latter is the characteristic peak. For V(V), there is no obvious peak at 630 or 765 nm, except for the strong adsorption band from 400 to 500 nm. As shown in Fig. 4e, the V in the pristine electrode and fully discharged sample (2.5D) are both tetravalent. Upon charging the sample to 3.8 V (3.8C), the intensity of the maximum peak at 765 nm declined to almost half of that of the pristine electrode. Moreover, there is no peak at 765 nm for the fully charged electrode, indicating complete conversion from tetravalent to pentavalent. In the case of the discharging process, the peak almost always returns to the original position and intensity, except for possible testing errors. Raman spectra were also used to acquire insight into the electrode structure states to confirm the stability of the NVOPF compound. As presented in Supplementary Fig. 24a, the band ranging from 800 to 1100 cm$^{-1}$ arises, owing to the stretching vibrations of PO$_4^{3-}$, whereas 200–550 cm$^{-1}$ corresponds to the bending vibrations of PO$_4^{3-}$. The band below 200 cm$^{-1}$ is attributed to the conversion of Na and V, and the vibration of the phosphate group[39]. During the entire charge–discharge process, there is no

obvious deviation or change in these characteristic bands, which demonstrates the robustness of the polyanionic structure. Moreover, the relative intensity ratio, $I_D/I_G$, gradually increases on charging and decreases upon discharging, as shown in Supplementary Fig. 24b. All these results confirm the high structural reversibility of NVOPF/8%KB during cycling, which contributes significantly to its superior Na-storage performance.

**Electrochemical performance in 26,650 prototypes vs. HC**. The excellent electrochemical properties and facile synthesis of NVOPF/KB motivated the large-scale preparation to demonstrate their feasibility of practical application. Figure 5a shows the charge–discharge curves of NVOPF/KB cathode and HC anode in half cell, respectively. The inset is the digital photos of 2 kg of mass-produced NVOPF/KB products and the cathode slurry. Supplementary Fig. 25 shows the XRD patterns and SEM image of the large-scale prepared product, which are consistent with the minor sample. The assembled prototype cells, 26,650 cylindrical batteries, were shown in Fig. 5b. They can deliver reversible capacities of 1502, 1405, 1398, 1375, 1349, and 1329 mAh at rates of 0.1, 0.2, 0.5, 1, 2, and 5C, respectively, as presented in Fig. 5c. The corresponding power density can be maintained at 94.5% over 100 cycles at 3C, as seen in Fig. 5d. However, how to optimize and improve the full cell to obtain a higher energy density is still going on.

## Discussion

In summary, a rapid and simple HEBM solid-state protocol for the fabrication of ultrafine polyanionic nanoparticles is proposed for the first time. In addition, an in situ NVOPF/KB nanocomposite has been constructed adopting the same strategy. The syntheses of NVPFs and NVOPF/KB are independent on the multiple vanadium source precursors. Interestingly, NVOPF/KB

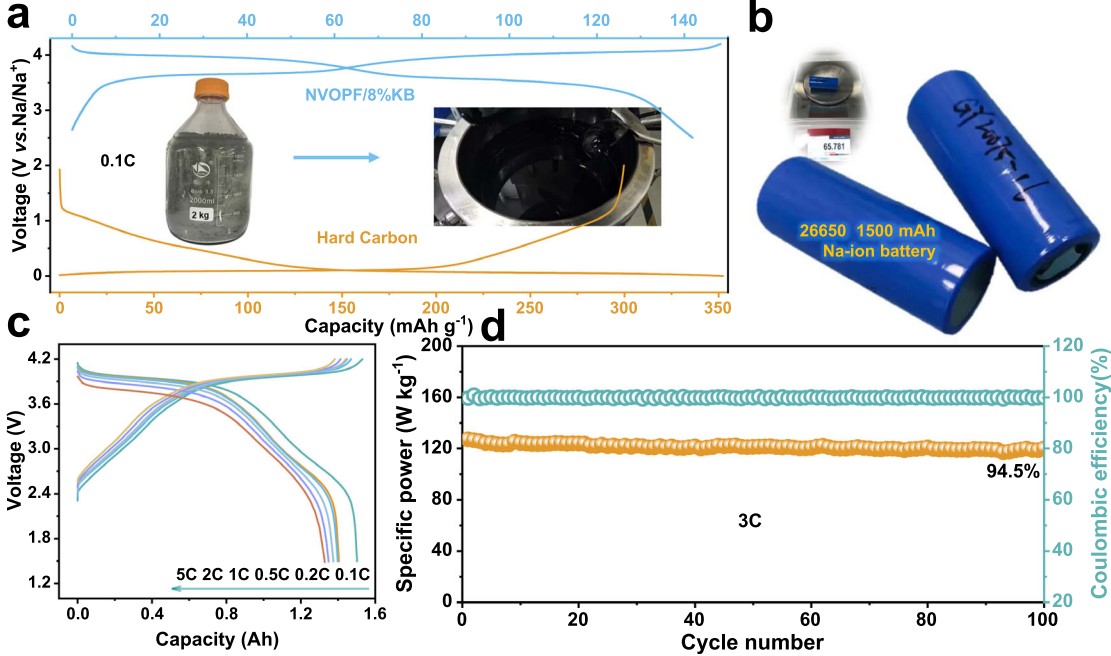

**Fig. 5 Demonstration of large-scale production of NVOPF/KB composite and prototype 26,650 full-cell performance. a** The charge–discharge profiles of mass-manufactured NVOPF/KB cathode and commercial hard carbon anode. Inset: the as-synthesized kilogram-scale NVOPF/8% KB products (left) and the preparation of the slurry (right). **b** Photograph of 26,650 prototype cells. Inset: the weight of a prototype 26,650 battery. **c** Rate capabilities of 26,650 cell from 0.1 to 5C. **d** Cycling performance of 26,650 cell at 3C.

nanocomposite can supply extra Na-storage sites. It also presents a superior rate capability with 113 mAh g$^{-1}$ at 20C and an ultralong cycling stability with 98% retention over 10,000 cycles, which is far beyond that of the bare NVOPF nanoparticles with 84.8 mAh g$^{-1}$ discharge capacity and 72% of capacity retention under the same conditions. To demonstrate the practical application of NVOPF/KB by HEBM, a scalable synthetic experiment was designed with 2 kg of the final product and the assembled 26,650 prototype cells were displayed. The optimization of full-cell performance still has a way to go by coupling advanced anode materials and optimized electrolyte. This is the first report about the synthesis of NVPFs by a HEBM strategy, which could be extended to synthesize other polyanion compounds.

## Methods

**Materials.** V$_2$O$_3$ (Sigma-Aldrich, 98%), VCl$_3$ (Sigma-Aldrich, 97%), V(acac)$_3$ (Sigma-Aldrich, 97%), VO$_2$ (Sigma-Aldrich, 99.9%), VOSO$_4$·xH$_2$O (Sigma-Aldrich, 97%), VO(acac)$_2$ (Sigma-Aldrich, 97%), V$_2$O$_5$ (Sinopharm, 99.5%), NH$_4$VO$_3$ (Aladdin, 99.0%), NaVO$_3$ (Aladdin, 99.0%), and Na$_3$VO$_4$·12H$_2$O (Aladdin, 99.0%) were used as vanadium sources in this work. H$_3$PO$_4$ (85%) and NaH$_2$PO$_4$·2H$_2$O (≥99%), Hydroxylamine (HONH$_2$·HCl, 99.5%) and (NaF, ≥98%) were purchased from Beijing Chemical Reagent Co. with analytical grade. Other reagents used for the preparation of coin cells were Na (Sinopharm, 99.8%), NaClO$_4$ (Alfa Aesar, 99%), fluoroethylene carbonate (FEC; Alfa Aesar, 99%), glass fiber separator (Whatman), KB (Lion, Carbon ECP600JD), and Acetylene black (AB; Alfa Aesar, ≥99.9%). The HC was purchased from KURARAY Co., Ltd, Japan. All reagents were used as received.

**Facile synthesis of NVPFs and NVOPF/KB.** In a typical synthesis of NVPFs particles, a certain amount of vanadium source, phosphorus, and fluorine source was employed to synthesize NVPFs, as listed in Supplementary Table 1. Hydroxylamine was added as a reductant to guarantee the complete transformation of the pentavalent vanadium sources to tetravalent. For example, 2 mmol NaVO$_3$, 3 mmol NaH$_2$PO$_4$·2H$_2$O, 2 mmol NaF, and 6 mmol HONH$_2$·HCl can be selected as precursors. Then, the mixture and several 5 mm stainless-steel milling balls (weight ratio of 1:10) were transferred to a three-dimensional ball-milling machine and were ball-milled at 600 r.p.m. for 30 min. For the construction of NVOPF/KB nanocomposite, different amount of KB was added and optimized. The designed weight of KB was based on the NVOPF with a V conversion rate of 94% in the synthetic process. After the reaction, the products were collected and then washed

several times alternately with distilled water and ethanol. The precipitates were dried in an oven at 120 °C for 10 h.

**Materials characterizations.** To characterize the purity and crystallinity of these materials, X-ray powder diffractions (XRDs) were conducted on an X-ray powder diffractometer (Bruker, D8 Advance) using Cu-Kα1 radiation (1.5406 Å) at 40 kV and 40 mA. The Rietveld structural refinement was conducted using the GSAS-EXPGUI software package[60]. The morphologies and sizes of the as-prepared samples were observed with a field-emission SEM (JEOL, JSM-7800FPRIME) equipped with an energy-dispersive X-ray spectroscope. HRTEM images were acquired on a JEM-2100F system. The BET surface area was deduced from a nitrogen adsorption isotherm analysis (Quantachrome, AUTOSORB-IQ-XR-C). The pH values were obtained using a pH meter (Hanna, HI 2221). The concentration of Na, V, and P in the aqueous phase was characterized by inductively coupled plasma atomic emission spectrometry (Perkin-Elmer, Optima 7000DV). The UV-vis spectrophotometer (Hitachi, U-4100) was used to record the UV-Vis spectra. The accurate content of KB in the composite of NVOPF/KB was measured by CHNS Elemental-analyser (Elementar, Vario Macro Cube). Raman spectra were measured using a Renishaw in Via Raman Microscope with a laser of 473 nm (HORIBA, labRAM HR evolution). Ex situ solid-state $^{23}$Na NMR spectra were acquired on a Bruker AVANCE III 400 spectrometer, at a $^{23}$Na Larmor frequency of 79.4 MHz. XPS data were collected by using ESCALAB 250Xi (Thermo Fisher Scientific monochromatized Mg/Al Kα radiation).

**Electrochemical measurements.** For NVOPF samples, the working electrodes were prepared by mixing the active materials, AB, and polytetrafluoroethylene binder at a mass ratio of 7:2:1. The composites were ground to square thin slices with a width of 0.9 cm and an active loading amount of 6–7 mg cm$^{-2}$, and then dried at 120 °C under vacuum for 6 h. All the specific capacities were calculated based on the weight of active material excluding KB. Coin-type (CR2032) cells were assembled in an argon-filled glove box (Unilab, M-BRAUN). All the electrochemical tests of half-cells were conducted on a Neware battery cycler (CT-4008T-5V10mA-164, Shenzhen, China) or Land CT2001A battery test system (Wuhan, China) at room temperature (25 °C) within a voltage window of 2.5–4.2 V. For the assembly of full cells, the fresh HC attached on Al foil was adopted as the anode and matched with NVOPF and NVOPF/8%KB, respectively. The capacity balance between the positive and negative electrodes was fixed as 1:1.2. The full cells were measured in the voltage range from 2.0 to 4.2 V, to make full use of the slope capacity of HC. NaClO$_4$ electrolyte (1.0 M) in a PC solution with 2 wt% FEC was used as the electrolyte. The counter electrodes were pure sodium foils. A thin sheet of microporous glass fiber (Whatman GF/D) served as the separator. CV performances were tested using an electrochemical workstation (Chenhua, CHI660C). The GITT measurement was programmed by supplying a constant current rate of 0.083C (1C = 130 mA g$^{-1}$) for 10 min followed by an open circuit stand for 40

min. The electrochemical impedance spectroscopy measurements were carried out at 25 °C on the Zahner IM6 in the frequency range of 1 to 10 MHz with an AC voltage.

## Data availability

The data that support the plots within these paper and other findings of this study are available from the corresponding author on reasonable request. Source data are provided with this paper.

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

## Acknowledgements

This work was supported by the National Natural Science Foundation of China (51872289), the DNL Cooperation Fund, CAS (DNL201914), the National Key Technologies R&D Program, China (2016YFB0901500), the Strategic Priority Research Program of the Chinese Academy of Sciences (XDA21070500), and Innovation Academy for Green Manufacture, CAS (Grant number IAGM2020C07) .

## Author contributions

J.Z. designed this work. X.S. performed the synthesis and characterizations. X.S. and M. H. carried out the electrochemical experiments and performed the lab in situ XRD. Q.Z. and X.Q. fabricated the 26,650 prototype cell and tested the electrochemical performance. B.L. helped with material synthesis. Q.Z. performed the X-ray photoelectron spectroscopy test. X.S., J.Z., and Y.-S.H. wrote the paper. C.Y. and H.L. revised the paper. All authors participated in analysis of the experimental data and discussions of the results, as well as in preparing the paper.

## Competing interests

The authors declare no competing interests.
