## [Peer Review File · Nature Communications]

Reviewers' comments:

Reviewer #1 (Remarks to the Author):

This work successfully synthesized NVPF compound and its carbon composite by a mechanochemical method using high energy ball milling. The developed NVPF/KB composite exhibits excellent electrochemical performance, including high specific capacity, rate capability, and cycling stability. In addition, they constructed a full cell of NVPF/KB cathode and hard carbon anode with Na₂C₄O₄ additive. The full cell shows a good cycling stability. While the electrochemical performance of the developed NVPF/KB cathode is excellent, this reviewer cannot find a significant novelty in their approach. The mechanochemical method has been used for the synthesis of various electrode materials for rechargeable batteries. In addition, it is hard to agree with that mechanochemical method using high energy ball milling is scalable to industrial levels. In fact, many battery industry people are very skeptical to use high energy ball milling for commercialization. In this respect, this reviewer feels that this manuscript is not suitable for a high standard journal like Nature Communications.

Here are minor points:

(i) It is really hard to read scale in Figure 2c-d.

(ii) Can the authors explain why the carbon composite delivers higher capacity than theoretical value? Does the high energy ball milling with carbon introduce defects in NVPF for extra capacity? Given extra capacity comes from low voltage (Figure S8a), the extra capacity might result from defects or amorphized structure/material.

Reviewer #2 (Remarks to the Author):

This manuscript communicates the mechanochemical synthesis of NVPFs as a cathode material for SIB with Ketjen Black as a carbon source. The cathode material was synthesized by using different precursors to investigate a potential inexpensive precursor with different reaction times (5, 15, and 30 minutes). The performance of the resultant electrode was quite impressive. The synthesis process is simple and rapid with superior electrochemical performance. The material characterization seems to be enough to support their data. My overall view of the paper is that while the work is of potential interest for publication after the following revision.

The detailed comments are listed below.

1. The mechanism of mechanochemical synthesis should be presented in detail. During the mechanochemical synthesis, what are the parameters followed? Why the mechanochemical synthetic method could produce highly crystalline and pure NVPFs? All the impurities or unreacted precursor are soluble in water? How did the impurities and unreacted precursor were removed?
2. Please explain reaction mechanism of hydroxylamine, which is used as reducing agent for vanadium.
3. Why the introduction of small amount of Ketjen Black (KB) can reduce the size of NVPF? The SEM images shows the agglomeration of particles with increase in percentage of KB. Secondly, if KB is

reducing the size, then why there is increase in size when you increase the percentage from 8%?

4. The explanation of “extra capacity” is very poor. Why KB add on extra sodium storage sites ? Please check charge and discharge capacity for KB to ensure that KB acts as storage sites for sodium. In addition, the active material loading in the composite electrode should be varied and tested electrochemical performance. In the experiment section, the active material loading should be presented. This is particularly important to examine the possibility of “extra capacity” because in some cases, the “extra capacity” comes from error in the active material loading.

5. In Fig. 3e, at a scan rate of 0.5 mV/s, the broad peak at near 3.4 V divided into two peaks. Why ?

6. In Fig 5e, in which state the Nyquist plots were gathered? Prior to cycling or after several cycling? If it was taken after several cycles, how about the EIS for prior to cycling state?

7. The English is very poor. For example, “Fig. S3. The results show that the best ratio of V: P: F is 1: 1.5: 1 owing to the best crystallinity and a high yield of 94%.” The whole manuscript should be carefully revised by a native English speaker.

Reviewer #4 (Remarks to the Author):

This work of Shen and co-workers investigates the synthesis of polyanionic cathode with a good performance for Na ion batteries

The authors present a very detailed and well-researched study on an important class of materials. There is also an usually large variety of complementary techniques used, and each method is difficult to perform. In terms of energy density, obtaining such a amount of $W h kg^{-1}$ is the key point for future development of na ion batteries. However, for industrial point of view more test should be necessary to afford its commercialization (then the energy density will be lowered).

Through characterization data are provided and the results are well aligned with previous reports in literature.

This cathode has two principal voltage reactions around 3.5 and 4.1 V. There are many reports in literature dealing with sodium vanadium phosphates and sodium vanadium fluorophosphates which exhibit the same performance. In my opinion the authors don't provide new data to be considered as of high impact to be published. Also, the full cell performance is very similar to that with compounds hard carbon / 1M NaClO₄ in PC+(FEC) / Na₃V₂(PO₄)₂F₃.

Overall, this is a well-designed study and a well-written manuscript suitable for publication in a high-impact journal, but does not reach the level required for Nat. Commun. in terms of novelty or groundbreaking conclusions or wide-reaching impacts.

Thank you for the reviewers' comments concerning our manuscript (NCOMMS-20-26882). Those comments are extremely valuable and helpful for revising and improving our paper, as well as the important guiding significance to our research. We have studied the comments carefully and have made corresponding corrections. We have revised the MS as suggested. The changes made in the manuscript have been highlighted in red color in the revised version and all the points raised by the reviewers have been addressed point by point. Please check the revised MS as well as our reply to the referees in the following page.

Our itemized responses to the reviewers' questions, comments and suggestions are as follows:

Reviewers' comments:

Reviewer #1 (Remarks to the Author):

This work successfully synthesized NVPF compound and its carbon composite by a mechanochemical method using high energy ball milling. The developed NVPF/KB composite exhibits excellent electrochemical performance, including high specific capacity, rate capability, and cycling stability. In addition, they constructed a full cell of NVPF/KB cathode and hard carbon anode with $\text{Na}_2\text{C}_4\text{O}_4$ additive. The full cell shows a good cycling stability. While the electrochemical performance of the developed NVPF/KB cathode is excellent, this reviewer cannot find a significant novelty in their approach. The mechanochemical method has been used for the synthesis of various electrode materials for rechargeable batteries. In addition, it is hard to agree with that mechanochemical method using high energy ball milling is scalable to industrial levels. In fact, many battery industry people are very skeptical to use high energy ball milling for commercialization. In this respect, this reviewer feels that this manuscript is not suitable for a high standard journal like Nature Communications.

Reply: Thank you for your comments. As we know, for many electrode materials, mechanochemical method has been widely used as an assistant procedure in the

synthetic process, which is only for precursors, not for target products, such as the precursors for LiFePO_4^1 , LiCoO_2^2 , $\text{LiMn}_2\text{O}_4^3$, etc. This work is the first attempt for the mechanochemical synthesis of polyanionic cathode for Na ion batteries. Besides, Reviewer #1 just suspected that it is hard to use mechanochemical method (high energy ball milling) in mass production. To demonstrate the practical application for the mechanochemical synthesis for NVOPF/KB composites, we have scaled up experiments to prepare 2 kg of product. In addition, we used them to assemble 26650 prototype cells with hard carbon as anode, as shown in the revised Figure 5.

Here are minor points:

(i) It is really hard to read scale in Figure 2c-d.

Reply: *Thanks for your kind reminder.*

We have carefully checked the scale marker for Figure 2c-d and redrew them in the revised manuscript.

(ii) Can the authors explain why the carbon composite delivers higher capacity than theoretical value? Does the high energy ball milling with carbon introduce defects in NVPF for extra capacity? Given extra capacity comes from low voltage (Figure S8a), the extra capacity might result from defects or amorphized structure/material.

Reply: *Thanks for your question.*

*As seen in Fig. S9, the excess capacity could derive from two reasons: 1) As you can see, from open-circuit voltage discharged to 2.5 V, the as-prepared NVOPF/KBs could release a capacity of 8.0 mAh g^{-1} (Fig. S9b). This part can be ascribed to the extra Na-storage sites in the NVOPF/KB interface, and this electrochemical behavior was also detected in nano- LiFePO_4^4 ; 2) The second part for the excess capacity can be contributed by KB itself. We have tested the electrochemical performance of pure KB and it has a discharge capacity of 12 mAh g^{-1} (Fig. S9a), which is calculated as $12 * 8 / (92 + 8) = 0.96 \approx 1 \text{ mAh g}^{-1}$ for KB's contribution. These two parts can contribute a total capacity of 9.0 mAh g^{-1} , which is consistent with the excess capacity.*

With respect to the defects in NVPF, it may exist in the NVPF. We have collected the

powder XRD pattern and Rietveld-refined results are shown in Fig. S15 and Table S9. The unit cell parameters are $a = b = 6.3896 \text{ \AA}$ and $c = 10.6601 \text{ \AA}$ with good reliability factor values ($R_{wp} = 6.37\%$, $R_p = 8.37\%$) for NVOPF/KB, which presents a larger lattice volume of 435.22 \AA^3 than NVOPF (434.98 \AA^3). Thus, the in-situ reaction in the ball milling process may produce defects or amorphized structure/material in the interface between NVOPF and KB, which could also contribute to a higher discharge capacity.

Reviewer #2 (Remarks to the Author):

This manuscript communicates the mechanochemical synthesis of NVPFs as a cathode material for SIB with Ketjen Black as a carbon source. The cathode material was synthesized by using different precursors to investigate a potential inexpensive precursor with different reaction times (5, 15, and 30 minutes). The performance of the resultant electrode was quite impressive. The synthesis process is simple and rapid with superior electrochemical performance. The material characterization seems to be enough to support their data. My overall view of the paper is that while the work is of potential interest for publication after the following revision.

Reply: *Thank you so much for your positive comment.*

The detailed comments are listed below.

1. The mechanism of mechanochemical synthesis should be presented in detail. During the mechanochemical synthesis, what are the parameters followed? Why the mechanochemical synthetic method could produce highly crystalline and pure NVPFs? All the impurities or unreacted precursor are soluble in water? How did the impurities and unreacted precursor were removed?

Reply: *Thanks for your good comments.*

The mechanism of mechanochemical synthesis does require more details. The specific raw material ratios were listed in Table S1. For better understanding, we have added the detailed synthetic parameters of NaVO_3 as vanadium source in the section of 'Methods' in the manuscript.

Mechanochemical method has been developed as a high-energy synthetic process in recent years^{5,6}. Compared with traditional synthesis method, it produces new materials by using mechanical energy to induce chemical reactions or structure changes. Mechanochemical strategy could significantly reduce the reaction activation energy, refine the grain, and enhance the combination of the interface and substrate to achieve a low-temperature chemical reaction^{7,8}. In fact, the mechanochemical reaction is equivalent to a kind of high-salt medium reaction, and it is much simpler than a co-precipitation method concerning reaction time and space.

After the high-energy ball milling reaction, all the impurities or unreacted precursors can be washed by distilled water or ethanol, ensuring the purity of as-prepared NVPFs.

Not all impurities or unreacted precursors are soluble in aqueous environment, such as the organic vanadium source including $V(acac)_3$, $VO(acac)_2$. They can be removed by ethanol. Although the vanadium oxide cannot dissolve in water, the addition of H_3PO_4 can help the dissolution of vanadium oxide in water. What's more, we have adjusted the dosage ratio of other precursors (which are soluble in water) to make full use of the high-value V raw materials. All of these guarantee to obtain the pure NVPFs.

2. Please explain reaction mechanism of hydroxylamine, which is used as reducing agent for vanadium.

Reply: *Thanks for your good question. Hydroxylamine is acted as reducing agent for vanadium. It has been reported that high-valence V^{5+} can be reduced to V^{4+} by hydroxylamine, especially under the acid conditions. The reaction mechanism can be written as follows^{9,10}:*

3. Why the introduction of small amount of Ketjen Black (KB) can reduce the size of NVPF? The SEM images shows the agglomeration of particles with increase in percentage of KB. Secondly, if KB is reducing the size, then why there is increase in

size when you increase the percentage from 8%?

Reply: *Thanks for your good question. After careful consideration, probably, the effect of size might not be significant enough to support the varies of discharge capacities, thus we removed the explanation of SEM images with increasing KB amount. As we know, KB is a type of conductive carbon with unique branch-chain structure, which shows a large specific surface area of $1337 \text{ m}^2/\text{g}^{11}$. The large surface area could undoubtedly contribute to superior electrical conductivity to construct a well-organized structure, leading to a better Na-storage performance. While excess amount of KB could cause severe aggregation between the particles to reduce the utilization of active materials, thus result in a decrease in discharge capacity.*

4. The explanation of “extra capacity” is very poor. Why KB add on extra sodium storage sites? Please check charge and discharge capacity for KB to ensure that KB acts as storage sites for sodium. In addition, the active material loading in the composite electrode should be varied and tested electrochemical performance. In the experiment section, the active material loading should be presented. This is particularly important to examine the possibility of “extra capacity” because in some cases, the “extra capacity” comes from error in the active material loading.

Reply: *Thanks for your good suggestion. The excess capacity can be divided into two parts: 1) As you can see, from open-circuit voltage discharged to 2.5 V, the as-prepared NVOPF/KBs could release a capacity of 8.0 mAh g^{-1} , this part can be ascribed to the extra Na-storage sites in the NVOPF/KB interface (Fig. S9b); 2) The second part for the excess capacity can be contributed by KB itself. We have tested the electrochemical performance of pure KB and found it could display a 12 mAh g^{-1} discharge capacity (Fig. S9a), which could calculated as $12*8/(92+8)=0.96\approx 1 \text{ mAh g}^{-1}$ for KB's contribution; These two parts could release a total contribution of 9.0 mAh g^{-1} capacity, which is well consistent with the excess capacity. Besides, the in-situ reaction in the ball milling process may produce defects or amorphized structure/material in the interface between NVOPF and KB, which could also contribute to a higher discharge capacity.*

Yes, active material loading amount is an important parameter for the calculation of capacity. We have double checked the loading amount and present the loading amount in the Electrochemical measurements part. The loading amount is 6-7 mg cm⁻². Besides, we confirm that the excess-capacity behavior is not caused by experimental errors.

5. In Fig. 3e, at a scan rate of 0.5 mV/s, the broad peak at near 3.4 V divided into two peaks. Why?

Reply: *Thanks for your good question. Compared with bare NVOPF, NVOPF/8%KB exhibits higher current response and the distinct reduction peak below 3.5 V was split into two mild peaks at 3.25 V and 3.46 V, which signifies the induced insertion of Na⁺ ion into two different crystallographic sites. The similar phenomenon has also been reported in other phosphate compounds, such as Na₃V₂(PO₄)₃¹², Na_{3.5}Mn_{0.5}V_{1.5}(PO₄)₃¹³. Besides, the introduction of KB could offer extra Na sites in the interface of NVOPF and KB, which also contributes to the division of the peak at 3.4 V.*

6. In Fig 5e, in which state the Nyquist plots were gathered? Prior to cycling or after several cycling? If it was taken after several cycles, how about the EIS for prior to cycling state?

Reply: *Thanks for your good question.*

Considering the confusion of sodiation agent, we have removed the related content from the original manuscript. The current manuscript presents the Nyquist plots in different states for half cells using bare NVOPF and NVOPF/8%KB cathodes, respectively, as seen in Fig. S21 and Table S11. As we can see, the Nyquist plots prior to cycling state shows the similar tendency compared to the charged cells. The half cell using NVOPF/8%KB as cathode has a lower charge transfer resistance than bare NOVPF, no matter for cells prior to cycling (245.3 Ω vs. 294.4 Ω) or after one cycle (137.3 Ω vs. 147.3 Ω), which further demonstrates the superior Na diffusion kinetics through the in-situ construction of the NVOPF/KB nanocomposite.

7. The English is very poor. For example, “Fig. S3. The results show that the best ratio of V: P: F is 1: 1.5: 1 owing to the best crystallinity and a high yield of 94%.” The whole manuscript should be carefully revised by a native English speaker.

Reply: *Thanks for your kind reminder. According to your suggestion, we have double checked the manuscript and revised it in detail.*

Reviewer #4 (Remarks to the Author):

This work of Shen and co-workers investigates the synthesis of polyanionic cathode with a good performance for Na ion batteries

The authors present a very detailed and well-researched study on an important class of materials. There is also an usually large variety of complementary techniques used, and each method is difficult to perform. In terms of energy density, obtaining such a amount of $Wh\ kg^{-1}$ is the key point for future development of na ion batteries. However, for industrial point of view, more test should be necessary to afford its commercialization (then the energy density will be lowered). Through characterization data are provided and the results are well aligned with previous reports in literature. This cathode has two principal voltage reactions around 3.5 and 4.1 V. There are many reports in literature dealing with sodium vanadium phosphates and sodium vanadium fluorophosphates which exhibit the same performance. In my opinion the authors don't provide new data to be considered as of high impact to be published. Also, the full cell performance is very similar to that with compounds hard carbon/1M NaClO₄ in PC+(FEC)/ Na₃V₂(PO₄)₂F₃.

Overall, this is a well-designed study and a well-written manuscript suitable for publication in a high-impact journal, but does not reach the level required for Nat. Commun. in terms of novelty or groundbreaking conclusions or wide-reaching impacts.

Reply: *Thanks for your comment. Reviewer #4 posed one question that it is hard to to perform the complementary techniques in the practical assembly process of storage batteries based on the dual consideration of technical problems and energy density. We accepted this suspect and removed the content related with sodiation agent from*

this work. Besides, to demonstrate the feasibility of practical application of NVOPF/KB composite, a scale-up experiment with 2 kg of product was conducted. Furthermore, 26650 prototype cells were fabricated. The corresponding results and discussion were presented in the newly added section: *Electrochemical Performance in 26650 Prototypes vs. Hard Carbon*.

Reviewer #4 also pointed that we didn't provide new data for NVPFs, and the full cell performance was very similar to $\text{Na}_3(\text{VPO}_4)_2\text{F}_3$. Firstly, we offer a novel strategy for the rapid synthesis for polyanionic cathodes with high rate and ultralong stability. Besides, to confirm the novelty and advantages of the current work, we have compared the synthetic processes and electrochemical performances with the reported state-of-art of NVPFs, and it was summarized in Figure S12c and Table S8. To the best of our knowledge, the tested electrochemical performances of our sample is the best, and the HC// $\text{Na}_3\text{V}_2(\text{PO}_4)_2\text{F}_3$ /KB prototype cells with an energy density of 88 Wh kg^{-1} were exhibited, which is superior to the reported HC// $\text{Na}_3\text{V}_2(\text{PO}_4)_2\text{F}_3$ full cell (75 Wh kg^{-1})¹⁴. The optimization of full-cell performance still has a way to go by coupling advanced anode materials and an even better electrolyte.

References

- 1 Shi, Z. *et al.* Synthesis of LiFePO_4/C cathode material from ferric oxide and organic lithium salts. *Electrochim. Acta* **56**, 4263-4267 (2011).
- 2 Zhang, J.-N. *et al.* Trace doping of multiple elements enables stable battery cycling of LiCoO_2 at 4.6 V. *Nat. Energy* **4**, 594-603 (2019).
- 3 Zhao, M. *et al.* Electrochemical Performance of Modified LiMn_2O_4 Used as Cathode Material for an Aqueous Rechargeable Lithium Battery. *Energy Fuel*. **26**, 1214-1219 (2012).
- 4 Duan, Y. *et al.* Excess Li-Ion Storage on Reconstructed Surfaces of Nanocrystals To Boost Battery Performance. *Nano Lett.* **17**, 6018-6026 (2017).
- 5 Amrute, A. P., Łodziana, Z., Schreyer, H., Weidenthaler, C. & Schüth, F. High-surface-area corundum by mechanochemically induced phase transformation of boehmite. *Science* **366**, 485 (2019).

- 6 Rao, R. P., Zhang, X., Phuah, K. C. & Adams, S. Mechanochemical synthesis of fast sodium ion conductor $\text{Na}_{11}\text{Sn}_2\text{PSe}_{12}$ enables first sodium–selenium all-solid-state battery. *J. Mater. Chem. A* **7**, 20790-20798 (2019).
- 7 Friščić, T., Mottillo, C. & Titi, H. M. Mechanochemistry for Synthesis. *Angew. Chem. Int. Edit.* **59**, 1018-1029 (2020).
- 8 James, S. L. *et al.* Mechanochemistry: opportunities for new and cleaner synthesis. *Chem. Soc. Rev.* **41**, 413-447 (2012).
- 9 Zhu, L. F. *et al.* Sodium metavanadate catalyzed one-step amination of benzene to aniline with hydroxylamine. *J. Catal.* **245**, 446-455 (2007).
- 10 Tang, D., Zhu, L., Qin, S., Su, Z. & Hu, C. A theoretical study on the mechanism of the oxidation of hydroxylamine by VO^{2+} . *J. Mol. Struct-theochem* **805**, 143-152 (2007).
- 11 Shen, X. *et al.* Controlled Synthesis of $\text{Na}_3(\text{VOPO}_4)_2\text{F}$ Cathodes with an Ultralong Cycling Performance. *Acs Appl. Energy Mater.* **2**, 7474-7482 (2019).
- 12 Lalère, F. *et al.* Improving the energy density of $\text{Na}_3\text{V}_2(\text{PO}_4)_3$ -based positive electrodes through V/Al substitution. *J. Mater. Chem. A* **3**, 16198-16205 (2015).
- 13 Zhang, J. *et al.* Understanding the superior sodium-ion storage in a novel $\text{Na}_{3.5}\text{Mn}_{0.5}\text{V}_{1.5}(\text{PO}_4)_3$ cathode. *Energy Storage Mater.* **23**, 25-34 (2019).
- 14 Broux, T. *et al.* High Rate Performance for Carbon-Coated $\text{Na}_3\text{V}_2(\text{PO}_4)_2\text{F}_3$ in Na-Ion Batteries. *Small Methods* **3**, 1800215 (2019).

Reviewer #1 (Remarks to the Author):

The authors claimed that mechano-chemical method has been used as an assistant procedure in the synthetic process, but not for target materials. However, this is not true. In fact, there are many examples that use “real” mechano-chemical reaction (without heat treatment) to synthesize electrode materials for Li and even Na-ion batteries. Examples are Na₂Fe₂O₅ (J. Mater. Chem. A 2020, 8, 20553), Disordered-Li₂MO₂F (Nature, 556, 185), NaMF₃ (J. Power Sources, 2009, 187, 247), Na₃MF₆ (Solid-State Ionics, 2012, 218, 35), LiFeSO₄F (Adv. Energy Mater. 2018, 8, 1701408).

For the scale up, it is impressive that the authors show the production of their material in 2-kg scale. However, as the authors might know, the practical synthesis of electrode materials is not in a few kgs-scale. While their demonstration is interesting and insightful, it does not indicate that their method is practical for mass (tons) production. But, the revision of this part and demonstration of kgs-scale synthesis is great.

Lastly, their argument on the extra-capacity from interface between NVPF/KB is really over-stated. They did not provide any direct evidences if the extra capacity is really coming from the interface. Of course, it is possible, but no evidence is provided.

Reviewer #2 (Remarks to the Author):

All the reviewer's concern is well responded.

Reviewer #3 (Remarks to the Author):

I think the authors have completed a good job after my comments. I think it can be published in the present form.

Regards

Dear Reviewers,

Thank you so much for reviewing our manuscript (NCOMMS-20-26882A-Z). We highly appreciate your professional comments on further improving the quality of our paper, as well as the important guiding significance to our research. We have considered the comments carefully and have made corresponding revision. The changes made in the manuscript have been highlighted in red color in the revised version and all the points raised by the reviewers have been addressed point by point.

The detail responses and the main corrections are summarized point-by-point as following:

Reviewers' comments:

Reviewer #1 (Remarks to the Author):

The authors claimed that mechano-chemical method has been used as an assistant procedure in the synthetic process, but not for target materials. However, this is not true. In fact, there are many examples that use “real” mechano-chemical reaction (without heat treatment) to synthesize electrode materials for Li and even Na-ion batteries. Examples are $\text{Na}_2\text{Fe}_2\text{OS}_2$ (J. Mater. Chem. A 2020, 8, 20553), Disordered- $\text{Li}_2\text{MO}_2\text{F}$ (Nature, 556, 185), NaMF_3 (J. Power Sources, 2009, 187, 247), Na_3MF_6 (Solid-State Ionics, 2012, 218, 35), LiFeSO_4F (Adv. Energy Mater. 2018, 8, 1701408).

For the scale up, it is impressive that the authors show the production of their material in 2-kg scale. However, as the authors might know, the practical synthesis of electrode materials is not in a few kgs-scale. While their demonstration is interesting and insightful, it does not indicate that their method is practical for mass (tons) production. But, the revision of this part and demonstration of kgs-scale synthesis is great.

Lastly, their argument on the extra-capacity from interface between NVPF/KB is really over-stated. They did not provide any direct evidences if the extra capacity is really coming from the interface. Of course, it is possible, but no evidence is provided.

Reply: Thank you for your comments. Frist, we appreciate your professional comments. We feel very sorry that we had a one-sided acknowledge about the mechano-chemical

method just being used as an assistant procedure in the synthetic process, but not for target materials due to the not-enough literature research. Therefore, we revised the description about the application of mechanochemical synthesis to prepare cathode materials in the part of introduction, and cited the related literatures.

As suggested, we have carefully read the literatures referred by the reviewer. These literatures confirmed that mechanochemical synthesis can be applied to the preparation of some cathode materials.¹⁻⁵ In contrast with their works, the current work exhibits the key innovations as follows: **1. It is the first time for the fabrication of sodium vanadium fluorophosphates via a mechanochemical protocol**, rather than the traditional high temperature solid-state method, hydrothermal or co-precipitation strategy. Though the mechanochemical synthesis can be utilized in some specific electrode materials, it is not always valid for all cathode materials. We proposed this method inspired by our previous reported work about the preparation of $\text{Na}_3(\text{VOPO}_4)_2\text{F}$ by room-temperature co-precipitation (**Joule 2, 2348, 2018**). In fact, mechanochemical synthesis can be regarded as a kind of co-precipitation method in the high-salt medium reaction system. **2. The current preparation of $\text{Na}_3(\text{VOPO}_4)_2\text{F}$ serves as an efficient and low-cost way with a rapid synthesis within 5-30 minutes and without atmosphere protection**, which can guarantee the easy scale-production. In our work, 2 kg of samples was kindly demonstrated. **3. Mechanochemical synthesis has been proved to be a well-designed process in this work, which integrates the reaction engineering, nanocrystallization and carbon-coating**. Just based on this special process, the synthesized $\text{Na}_3(\text{VOPO}_4)_2\text{F}/\text{KB}$ cathode exhibits an **interfacial Na-storage effect** and displays excess capacity beyond the theoretical value.

For the scale up, we thank you for your recognition. The kg-scale synthesis is the first step for the industrialization trial, we still have a long way to go for the desired mass (tons) production.

The reviewer also pointed that the explanation of **extra-capacity from interface** between NVPF/KB is poor. As far as we know, the extra-capacity behavior is widespread in some anode materials. Numerous reports have shown that the reversible capacity of various electrode materials can exceed their theoretical value based on the

conventional mechanisms. For example, the additional capacity was also reported in LiFePO_4 cathode material, such as *Nat. Commun.* **2013**, *4*, 1687; *Funct. Mater. Lett.* **2016**, *9*, 1650053; *J. Alloy. Compd.* **2020**, *835*, 155148; *Nano Energy*, **2017**, *34*, 408 and *ACS Nano*, **17**, 6018, etc.⁶⁻¹² In these studies, the extra-capacity in LiFePO_4 was ascribed to the additional Li-C interaction between the active material and different forms of carbon. For example, Guo et al. have proposed the similar excess lithium storage properties in LiFePO_4 -carbon interface by ball-milling.¹³ In this work, the ball-milling method was applied to the mixture of LiFePO_4 and carbon, and the effective capacity larger than the theoretical one by 30 mAh g^{-1} was achieved. Besides, X-ray photoelectron spectroscopy is used to investigate the surface of LiFePO_4 and carbon black. From the Li 1s XPS spectra, Li_xC signal was detected on the surface of LiFePO_4/C in the ball-milled sample; While for the non-milled sample, no Li_xC was observed. This result demonstrates that lithium ion could be stored onto the surface of carbon black, forming Li-C bond in the ball-milled process. They used X-ray photoelectron spectroscopy or theoretical calculation to explain the excess capacity without offering the intuitive evidence because the fuzzy boundary interface is hard to measure or define during the observations. This could be an inspiration to solve our problem.

Generally, the whole capacity can be divided into three parts: the active material contribution, the carbon contribution and the interface contribution between active material and carbon. Based on the above analysis, we firstly reconfirmed the discharge capacity using five batches of NVOPF/8%KB samples, and the average capacity was calculated to be 141.9 mAh g^{-1} (**Supplementary Fig. 9a**). Then, we investigated the Na-storage capacity for the discharged KB with and without ball-milling. It can be found that the discharged capacity of KB with and without ball-milling are 3.5 and 0.7 mAh g^{-1} , respectively, which proves the Na-storage of ball-milled KB itself (**Supplementary Fig. 9b-c**). The X-ray photoelectron spectroscopy was employed to detect the chemical state of discharged KB electrodes, as shown in **Supplementary Fig. 9d**. No Na signal was observed in the discharged KB without ball-milling, but an obvious Na 1s peak located at $\sim 1069.2 \text{ eV}$ can be found in the discharged KB with ball-milling, which

confirms the binding effect between Na and ball-milled KB. Based on this, we collected the Na 1s XPS data of NVOPF and NVOPF/8%KB electrodes with varied charge/discharge states. As shown in **Supplementary Fig. 10a**, there was no obvious change in the Na peak for NVOPF during the first charge-discharge process (first charged to 4.2 V, then discharged to 2.5 V), declaring only Na in the crystal structure can be migrated in the electrochemical reaction. Whereas for NVOPF/8%KB, the extra Na-C peak can be detected in the pristine electrode, implying that Na-C bond forms in the ball-milled process. After charged to 4.2 V, the Na-C peak disappeared as the extraction of Na⁺ from the NVOPF/8%KB species. When the NVOPF/8%KB electrode was discharged to 2.5 V, an extra Na peak emerged at 1071.7 eV, which denotes the formation of new Na-binding at this state, as seen from **Supplementary Fig. 10b**. The new Na-binding is not derived from the bulk of NVOPF or KB, but could be derived from the interfacial between NVOPF and KB. In 2003, Maier et al. proposed the nanocrystallinity effects in Li-ion materials and the **excess interfacial storage mechanism** named **charge separation at phase boundaries**.^{14,15} Inspired by the extra Li-storage, the current interfacial Na-storage can be illustrated in **Supplementary Fig. 11**. In the interface between Na₃(VOPO₄)₂F and KB, Na⁺ can be accommodated at the boundary of NVOPF side while the electrons are restricted to the KB side. For the combination of Na₃(VOPO₄)₂F and KB, the stored Na⁺ and e⁻ act as a bridge during the charge-discharge process.¹⁶ In this case, an interfacial Na-storage effect can be expected. To sum up, **the XPS results with varied states confirm that the ball-milled materials have the extra interfacial Na-storage behavior.**

Supplementary Fig. 9 (a) The first-discharge specific capacity of 5 batches of NVO/PF/8%KB cathode samples. The discharge curves of KB from OCV (b) without and (c) with ball-milling process at the current rate of 0.1 C. (d) XPS Na 1s spectra of discharged KB electrodes with and without ball-milling.

Supplementary Fig. 10 XPS Na 1s spectra of (a) NVOF and (b) NVOF/8%KB with varied charge-discharge states

Supplementary Fig. 11 The schematic diagram of NVOPF/8%KB interfacial Na-storage mechanism based on charge separation at phase boundaries

Reviewer #2 (Remarks to the Author):

All the reviewer's concern is well responded.

Reply: *Thank you for your positive comment.*

Reviewer #3 (Remarks to the Author):

I think the authors have completed a good job after my comments. I think it can be published in the present form.

Reply: *Thank you so much for your positive comment.*

Reference

- 1 Gamon, J. *et al.* Na₂Fe₂OS₂, a new earth abundant oxysulphide cathode material for Na-ion batteries. *J. Mater. Chem. A* **8**, 20553-20569 (2020).
- 2 Lee, J. *et al.* Reversible Mn²⁺/Mn⁴⁺ double redox in lithium-excess cathode materials. *Nature* **556**, 185-190 (2018).
- 3 Gocheva, I. D. *et al.* Mechanochemical synthesis of NaMF₃ (M=Fe, Mn, Ni) and their electrochemical properties as positive electrode materials for sodium batteries. *J. Power Sources* **187**, 247-252 (2009).
- 4 Shakoor, R. A. *et al.* Mechanochemical synthesis and electrochemical behavior of Na₃FeF₆ in sodium and lithium batteries. *Solid State Ionics*. **218**, 35-40 (2012).
- 5 Seo, D.-H. *et al.* Intrinsic nanodomains in triplite LiFeSO₄F and its implication in lithium-ion diffusion. *Adv. Energy Mater.* **8**, 1701408 (2018).
- 6 Duan, Y. *et al.* Excess Li-ion storage on reconstructed surfaces of nanocrystals to boost battery performance. *Nano Lett.* **17**, 6018-6026 (2017).
- 7 Li, Q. *et al.* Extra Li-ion storage of LiFePO₄/C composite materials synthesized with Fe_{1.5}P. *J. Alloy. Compd.* **835**, 155148 (2020).
- 8 Lung-Hao Hu, B., Wu, F.-Y., Lin, C.-T., Khlobystov, A. N. & Li, L.-J. Graphene-modified LiFePO₄ cathode for lithium ion battery beyond theoretical capacity. *Nat. Commun.* **4**, 2705 (2013).
- 9 Wang, J. *et al.* Interaction of carbon coating on LiFePO₄: A local visualization study of the influence of impurity phases. *Adv. Funct. Mater.* **23**, 806-814 (2013).
- 10 Varzi, A., Ramirez-Castro, C., Balducci, A. & Passerini, S. Performance and kinetics of LiFePO₄-carbon bi-material electrodes for hybrid devices: A comparative study between activated carbon and multi-walled carbon nanotubes. *J. Power Sources* **273**, 1016-1022 (2015).
- 11 Zhao, Q. *et al.* Phytic acid derived LiFePO₄ beyond theoretical capacity as high-energy density cathode for lithium ion battery. *Nano Energy* **34**, 408-420 (2017).
- 12 Ren, W. *et al.* Soft-contact conductive carbon enabling depolarization of LiFePO₄ cathodes to enhance both capacity and rate performances of lithium

- ion batteries. *J. Power Sources* **331**, 232-239 (2016).
- 13 Guo, H., Song, X., Zheng, J. & Pan, F. Excess lithium storage in LiFePO₄-carbon interface by ball-milling. *Funct. Mater. Lett.* **9**, 1650053 (2016).
 - 14 Jamnik, J. & Maier, J. Nanocrystallinity effects in lithium battery materials (Aspects of nano-ionics. Part IV). *Phys. Chem. Chem. Phys.* **5**, 5215-5220 (2003).
 - 15 Maier, J. Nanoionics: ion transport and electrochemical storage in confined systems. *Nat. Mater.* **4**, 805-815 (2005).
 - 16 Balaya, P. et al. Nano-ionics in the context of lithium batteries. *J. Power Sources* **159**, 171-178 (2006).

Reviewer #1 (Remarks to the Author):

Thank you for taking care of all the concerns. Now, they are well addressed in the revised manuscript and it is now ready to publication.

Responses to the Referees' Comments

Thank you so much for reviewing our manuscript (NCOMMS-20-26882B). We highly appreciate your positive comments.

Reviewer #1 (Remarks to the Author):

Thank you for taking care of all the concerns. Now, they are well addressed in the revised manuscript and it is now ready to publication.

Reply: *Thank you so much for your positive comment.*